# Food insecurity and COVID-19-related experiences among people with HIV: A mixed methods analysis and conceptual framework

Dini Harsono[1,2]۞*, Kelsey Sklar[3]۞, Michael Silver[4],
Mayange Frederick[5], Merceditas Villanueva[1,6], E. Jennifer Edelman[1,2],
Jessica E. Yager[1,5]

1 Center for Interdisciplinary Research on AIDS, Yale School of Public Health, New Haven, Connecticut, United States of America, 2 Section of General Internal Medicine, Yale School of Medicine, New Haven, Connecticut, United States of America, 3 Department of Emergency Medicine, SUNY Downstate Health Sciences University, Brooklyn, New York, United States of America, 4 Department of Epidemiology, School of Public Health, SUNY Downstate Health Sciences University, Brooklyn, New York, United States of America, 5 Division of Infectious Diseases, Department of Medicine, SUNY Downstate Health Sciences University, Brooklyn, New York, United States of America, 6 HIV/AIDS Program, Section of Infectious Disease, Department of Internal Medicine, Yale School of Medicine, New Haven, Connecticut, United States of America

۞ These authors contributed equally to this work and share first authorship.
* dini.harsono@yale.edu

## Abstract

Food insecurity is a key driver of health disparities in the United States and globally, and is associated with increased likelihood of chronic diseases and poorer health outcomes. Social and economic disruptions during the COVID-19 pandemic increased the level of food insecurity and disproportionately impacted low-income households and racial and ethnic minority groups. In this mixed methods study, we explored relationships between sociodemographic factors, HIV care engagement, and experiences of food insecurity in people with HIV (PWH) during the first wave of the COVID-19 pandemic. We administered a one-time telephone survey to PWH engaged in two HIV clinics in the U.S. Northeast (n = 283) and conducted four focus groups with clinical staff (n = 23). Among the surveyed PWH, 30.7% (n = 87) reported experiences of food insecurity early in the COVID-19 pandemic. Receiving care in Brooklyn, NY, being unemployed prior to the pandemic, and experiencing some or a lot of changes in daily routine due to the pandemic were associated with food insecurity experiences. Qualitative analysis of patients' free-text survey responses and clinical staff focus group transcripts identified factors at individual, intrapersonal, community, and structural levels contributing to food insecurity experiences during the pandemic. Informed by the findings, we developed a conceptual social-ecological framework illustrating the linkages between experiences of food insecurity and COVID-19 among PWH. Efforts to address food insecurity among PWH should include screening for food needs in HIV care settings, communicating about food

**Data availability statement:** The survey data and qualitative codebook associated with this manuscript have been submitted as supplementary materials. The authors confirm that the submitted survey dataset (deidentified) contains all raw data required to replicate the quantitative results of our study. The authors also submitted the codebook used in the qualitative analysis of the reported study. Due to the small sample size and potential for re-identification, the qualitative data collected in this study will not be made available for public sharing.

**Funding:** This work was supported by funding from the National Institute of Mental Health (grant #P30MH062294, P30 grant PI: Kershaw, pilot study grant PI: Edelman) and the National Cancer Institute (grant #R01CA243910, R01 grant PIs: Edelman and Bernstein) of the National Institutes of Health. The content is solely the responsibility of the authors and does not necessarily represent the official views of the Center for Interdisciplinary Research on AIDS or the National Institutes of Health. The funders had no role in study design, data collection and analysis, decision to publish, or preparation of the manuscript.

**Competing interests:** All authors have no competing interests to disclose that are relevant to the content of this paper.

assistance resources and programs, and implementing evidence-based interventions that can improve food security and nutrition.

## Introduction

Food insecurity, defined by the U.S. Department of Agriculture (USDA) as "household-level economic and social condition of limited or uncertain access to adequate food" [1], is a key driver of health disparities [2,3]. A USDA report shows that 47.4 million individuals lived in food-insecure households in the U.S. in 2023 [4]. The report also notes that while the average rate of household food insecurity in the U.S. was 13.5%, rates were significantly higher for households with non-Hispanic Black (23.3%) and Hispanic (21.9%) household owners or renters, and in female-headed households (34.7%) [4].

Food insecurity affects people with HIV (PWH) in both low- and high-income settings [5–7], including the United States [8,9], and contributes to lower engagement in care and poor HIV-related outcomes [8,10–12]. The relationship between food insecurity and HIV health outcomes appears to be bidirectional and complex with potential direct and indirect effects [8,13,14]. Food insecurity may increase the risk of HIV acquisition among HIV-negative individuals due to high-risk behaviors such as exchanging sex for money or non-monetary items [14]. In PWH, food insecurity could contribute to adverse health outcomes such as unsuppressed viral loads due to poor antiretroviral therapy (ART) adherence, as the need to obtain food might interfere with keeping medical appointments and taking medicines as prescribed [10,13,15]. In turn, a growing body of evidence demonstrates that HIV infection influences the likelihood of food insecurity. HIV/AIDS-related morbidity and mortality could cause physical debilitation among PWH that may lead to decreased economic productivity and household income, greater caregiver burden, and increased use of unhealthy coping strategies such as unprotected sex [11,13,16].

The COVID-19 pandemic caused social and economic hardship across communities and disrupted the health care systems, as it simultaneously magnified and exacerbated health disparities in the United States and globally. A number of studies show evidence of a heightened risk for acquisition of SARS-CoV-2, the virus that causes COVID-19, in individuals due to their occupational exposure [17,18], housing vulnerabilities [19], or reduced access to essential services [20,21]. Further, recent review papers have explored how the COVID-19 pandemic, the resulting public health measures, and social determinants of health can impact individuals' acquisition risk of SARS-CoV-2 and COVID-19 outcomes [22,23]. However, evidence on the mechanisms and degrees through which these factors may drive COVID-19 acquisition and social implications among PWH remains limited and unclear [24–27].

Given the intersecting effects of food insecurity and HIV, we sought to characterize the impact of the COVID-19 pandemic and the resulting public health control measures on food insecurity among PWH in two cities in the Northeastern U.S. during the region's first pandemic surge in 2020 [28]. To do so, we conducted a mixed methods

exploratory study to examine the relationships between sociodemographic factors, HIV care engagement, food insecurity experiences, and COVID-19-related experiences among PWH. Given the disruptions caused by the COVID-19 pandemic on food provision services and resources, this study has potential to inform future strategies to mitigate food insecurity among PWH and improve health outcomes.

## Materials and methods

### Study overview

Between May 15, 2020 and August 11, 2020, we conducted a mixed methods study engaging individuals with HIV and clinical staff to investigate the impact of the COVID-19 pandemic on PWH at two academic HIV clinics in large urban centers in the U.S. Northeast [29,30]. Patient participants completed a one-time telephone survey of close-ended and open-ended questions to evaluate their experiences with the COVID-19 pandemic. Additionally, we invited clinical staff to participate in virtual focus groups to gain insights from providers with diverse areas of expertise. The study protocol was approved by the institutional review boards (IRBs) at Yale University (#2000028033) and SUNY Downstate Health Sciences University (#1593449−1) and was HIPAA compliant.

### Settings, participants, and procedures

This study was conducted at two academically-affiliated HIV clinics located in New Haven, Connecticut and Brooklyn, New York, that provide extensive co-located wrap-around services to PWH. As the current study leveraged existing research collaborations focused on tobacco use among PWH [31,32], we actively recruited PWH who had documentation of current tobacco use in the electronic medical record (EMR). Patients without active tobacco use were recruited at the New Haven, CT clinic. Other inclusion criteria were: (1) ≥ 18 years of age; (2) English-speaking; and (3) able to give informed consent. Clinical staff (i.e., clinicians, staff, and clinical leadership) working at the two clinics were invited to participate in virtual focus groups. Due to the minimal risk involved and the social-distancing restrictions of the COVID-19 pandemic, all participants provided verbal informed consent for participation in research activities. Participants were provided a $30 gift card for study participation.

### Data collection

**Patient surveys.** The "COVID-19 and PWH Survey", previously published [29], was administered by telephone by trained research staff who followed the study procedures outlined in the IRB protocols; data were entered directly into the secure web-based Research Electronic Data Capture (REDCap) system [33,34]. Informed by an existing survey study led by Wolf and colleagues [35], the survey measured several areas, including sociodemographic characteristics, COVID-19 related awareness, concerns, actions, and impacts, and experiences with healthcare use during the pandemic [29].

*Outcomes of interest*. To measure the experiences of food insecurity, we examined responses to the following question: "How much has COVID-19 impacted your ability to get or pay for food?" Responses to this question were recorded in a four-point Likert scale ("not at all," "a little," "some," or "a lot"). We also assessed participants' use of food assistance services during the COVID-19 pandemic, indicated by patient participant selecting "I go to a food pantry/soup kitchen/food pick-up location for food" in response to the question: "What has made it hard for you to practice social distancing?". Free-text comments or additional explanations were captured verbatim by research staff conducting the phone survey.

*Independent variables*. Participant characteristics included sociodemographic characteristics (i.e., age, race, ethnicity, gender identity, household income, employment status, housing situation, number of people in household); HIV biomarkers based on most recent labs recorded in the EMR (i.e., CD4 cell count, presence of a detectable HIV viral load defined as >50 copies/mL); and engagement in HIV care (i.e., documentation of current ART prescription and at least one HIV clinic visit in the 6 months prior to survey date). Socioeconomic changes in the context of COVID-19 pandemic

were assessed with the following questions: 1. "How much has the coronavirus changed your daily routine?" (a four-point Likert scale: "not at all," "a little," "some," or "a lot"); 2. "Has there been any change in your employment status due to coronavirus?" (responses were recoded into three categories: "change in employment due to the pandemic" ("lost job", "furloughed", "not working for pay, but not fired", "pay cut", "reduction of hours", "increased hours", "got a new job outside of the home", "not working but still being paid", or "other"),"not employed prior to the pandemic", and"unchanged"); and 3. "Has COVID-19 impacted your housing situation?" (response options were "no, I am in the same place", "no, but I still don't have a regular place", "yes, I had to move, but found a place to stay", or "yes, I no longer have a place to stay"). Regarding COVID-19 exposure and testing experiences, we asked whether participants knew that they had or previously had SARS-CoV-2 (response options were "yes, currently have it", "yes, previously had it", "no", or "don't know"), and whether participants had ever been tested for SARS-CoV-2 (if yes, reasons for testing and testing outcome, and if no, reasons for not getting tested).

**Clinical staff focus groups.** Two study team members conducted a total of four focus groups, two at each clinic site, with staff and clinicians using Zoom© videoconferencing platform. Study team members at each site invited staff and clinicians via email to participate in the focus groups. Broad questions and follow-up probes were used to assess staff perceptions of the impact of the COVID-19 pandemic on patients and HIV care [29]. The mean duration of the focus groups was 55 minutes (ranging from 52 to 57 minutes). Focus groups were digitally recorded and transcribed by Zoom. Focus group transcripts were subsequently de-identified and reviewed for accuracy by a member of the research team, and stored in a secure server. Upon completion of each focus group, participants were asked to complete a brief REDCap-based survey regarding their demographics and clinic role.

## Data analyses

**Quantitative analysis: Patient surveys.** The primary outcome of food insecurity experiences was created by dichotomizing patients' responses to the survey question, "How much has the COVID-19 pandemic impacted your ability to get or pay for food?" Responses indicating "not at all" were compared with combined responses of "a little", "some", and "a lot". All numeric predictor variables were summarized with median and interquartile range (IQR) ($25^{th} – 75^{th}$ percentile) and compared across groups using the Wilcoxon rank sum test. All categorical predictor variables were summarized using frequency and percentage and compared across groups using a Fisher's exact test. We also stratified analyses by site, recognizing that characteristics of individuals experiencing food insecurity as well as available resources to mitigate its impact could vary by location.

A multivariate logistic regression model was fit to the data to determine predictors of and associations with food insecurity experiences. All predictors that were univariately associated with food insecurity experiences in the above comparison (defined as statistically significant at $p < 0.05$) were used as predictors in the multivariable model. We used listwise deletion to handle missing data. All data were analyzed using SAS version 9.4 software (Copyright© 2013, SAS Institute Inc., Cary, NC, USA). Open-ended responses to survey questions were analyzed using content analysis (described further below) [36].

**Qualitative analysis: Patients' free-text responses and clinical staff focus groups.** To analyze both free-text responses provided by patient participants and clinical staff focus groups, we used a rapid assessment process followed by an inductive process of iterative coding to identify relevant themes using content analysis [36,37]. Three members of the study team independently reviewed each focus group transcript line-by-line to develop and refine the codebook and reach consensus on codes. Themes were then generated based on coded quotations and consensus by four study team members, and refined following discussion with the broader study team.

## Data integration and triangulation

Analyses of survey data were conducted in parallel with qualitative focus group data. Qualitative themes developed based on analysis of focus group transcripts were then merged with themes identified from patients' free-text

responses to the relevant open-ended survey items. Triangulation was employed at various stages in order to establish consistency of data analysis and interpretation. Guided by Patton's triangulation approaches [38], we employed four types of triangulation. First, we conducted triangulations of methods and sources to establish themes and consistency between quantitative and qualitative data. Second, we employed investigator triangulation by involving multiple study team members to review findings and assess the accuracy of the data. Lastly, we applied the social-ecological model [39,40] to guide the theory triangulation of quantitative and qualitative data to consider the linkages between the individual, interpersonal, community and structural factors impacting food insecurity experiences in the context of the COVID-19 pandemic.

## Results

### Quantitative results

**Patient participant characteristics.** We outreached 719 individuals out of a random sample of 755 patients who met the eligibility criteria; 40.2% (n = 289) were not successfully contacted by telephone despite multiple attempts. A total of 283 PWH (65.8% of those who were successfully outreached) agreed to participate in the survey, yielding a response rate of 39.4%. Sociodemographic and clinical characteristics of patient participants are described in Table 1.

Of 283 patient participants, 104 (36.8%) were engaged in care in Brooklyn, NY and 179 (63.3%) were engaged in care in New Haven, CT (Table 1). The median age of participants was 55 years old, with 51.2% (n = 145) individuals identifying as male. No participants identified as transgender or non-binary, though two participants indicated a different current gender from their sex at birth. Almost two-thirds of participants (n = 185, 65.4%) identified as Black, and 84% (n = 237) identified as non-Hispanic. Among participants receiving care in Brooklyn, NY, 79.8% (n = 83) identified as Black, compared to 57% (n = 102) in New Haven, CT (p < 0.001). Only 7.7% (n = 8) of those who accessed care in Brooklyn, NY identified as White, compared to 31.3% (n = 56) in New Haven, CT (p < 0.001). The majority of participants (87%, n = 240) reported a household income below $50,000. Nearly all (n = 99, 97%) participants engaged in care in Brooklyn, NY had a household income below $50,000, compared to 82% (n = 141) of those in New Haven, CT (p < 0.001). The median household size among participants was two individuals (data not shown in Table 1). Almost half of the participants lived with a child (n = 129, 45.6%) and a quarter of participants (n = 74, 26.1%) lived with a spouse/partner. Participants engaged in care at the New Haven, CT site were more likely to live with a spouse/partner (30.7%, n = 55) than those at the Brooklyn, NY site (18.3%, n = 19) (p = 0.02).

Regarding clinical characteristics, most of surveyed patient participants were connected to health care: 88.1% (n = 244) had completed an HIV clinic visit in the past 6 months and 97.5% (n = 270) were prescribed ART. Most recent CD4 cell counts showed a median of 609 cells/mm$^3$, and 83% of participants (n = 236) had an undetectable HIV viral load from their last clinic visit. More patient participants engaged in care at the Brooklyn, NY site had detectable HIV viral load than those from the New Haven, CT site (29.1% vs. 9.8%, p < 0.001). Patient participants receiving care in New Haven, CT were more likely to have completed an HIV clinic visit in the past six months than patients who received care in Brooklyn, NY (97.1% vs. 72.8%, p < 0.001). There was no significant difference in ART prescription status or CD4 cell counts by site (p value >0.05).

**Patient participants' COVID-19 experiences.** During the COVID-19 pandemic, 87.2% (n = 246) reported experiencing changes in their daily routine, among whom nearly half (42.2%, n = 119) reported the pandemic had changed their routine a lot (Table 1). Regarding employment status, 16% (n = 44) of patient participants experienced changes in their employment status as a result of the COVID-19 pandemic, while more than half (59.6%, n = 164) had no change in employment. More patient participants engaged in care in New Haven, CT reported an unchanged employment status due to the COVID-19 pandemic (73.9% vs. 34.3%, p < 0.001), while participants in care in Brooklyn, NY were more likely to report being unemployed before the pandemic (52.5% vs. 8.5%, p < 0.001). A small number of participants (3.9%, n = 11) reported changes in their housing situation: either they moved and found a new place to stay (3.2%), were without

**Table 1. Participant Characteristics Overall and by Locations (n = 283).**

| Characteristic | | Total (n = 283)* | Brooklyn, NY (n = 104) | New Haven, CT (n = 179) | p-value |
|---|---|---|---|---|---|
| **Demographic characteristics** | | | | | |
| **Age**, years, median (IQR) | | 55 (45 - 60) | 55 (48 - 61) | 54 (45 - 60) | 0.2 |
| **Sex at birth**, n (%) | Male | 147 (51.9%) | 49 (47.1%) | 98 (54.7%) | 0.22 |
| | Female | 136 (48.1%) | 55 (52.9%) | 81 (45.3%) | |
| **Current gender**, n (%) | Male | 145 (51.2%) | 49 (47.1%) | 96 (53.6%) | 0.32 |
| | Female | 138 (48.8%) | 55 (52.9%) | 83 (46.4%) | |
| **Race**, n (%) | White/Caucasian | 64 (22.6%) | 8 (7.7%) | 56 (31.3%) | **<0.001** |
| | Black/African American | 185 (65.4%) | 83 (79.8%) | 102 (57%) | |
| | Native Hawaiian/ Pacific Islander | 1 (0.4%) | 0 (0%) | 1 (0.6%) | |
| | More than one race | 4 (1.4%) | 4 (3.8%) | 0 (0%) | |
| | Other | 28 (9.9%) | 9 (8.7%) | 19 (10.6%) | |
| | Refused | 1 (0.4%) | 0 (0%) | 1 (0.6%) | |
| **Ethnicity**, n (%) | Hispanic | 45 (16%) | 15 (14.4%) | 30 (16.9%) | 0.62 |
| | Non-Hispanic | 237 (84%) | 89 (85.6%) | 148 (83.1%) | |
| **Annual household income**, n (%) | $0-$49,999 | 240 (87%) | 99 (97%) | 141 (82%) | **<0.001** |
| | $50,000 or more | 35 (13%) | 3 (3%) | 32 (18%) | |
| **HIV-related variables§** | | | | | |
| **Detectable HIV viral load[a]**, n (%) | Yes | 47 (17%) | 30 (29.1%) | 17 (9.8%) | **<0.001** |
| **CD4 cell count,** cells/mm3, median (25th - 75th) | | 609 (395.2 - 927.6) | 654 (333 - 967) | 588 (407.8 - 880) | 0.94 |
| **Prescribed ART**, n (%) | Yes | 270 (97.5%) | 102 (99%) | 168 (96.6%) | 0.26 |
| **HIV clinic visit completion in past 6 months**, n (%) | Yes | 244 (88.1%) | 75 (72.8%) | 169 (97.1%) | **<0.001** |
| **Socioeconomic changes in the context of COVID-19 pandemic** | | | | | |
| **Food insecurity experiences[c]**, n (%) | Yes | 87 (30.7%) | 51 (49%) | 36 (20.1%) | **<0.001** |
| **How much has the coronavirus changed your daily routine?**, n (%) | Not at All | 36 (12.8%) | 13 (12.6%) | 23 (12.8%) | 0.61 |
| | A Little | 63 (22.3%) | 19 (18.4%) | 44 (24.6%) | |
| | Some | 64 (22.7%) | 23 (22.3%) | 41 (22.9%) | |
| | A Lot | 119 (42.2%) | 48 (46.6%) | 71 (39.7%) | |
| **Has there been any change in your employment status due to coronavirus?**, n (%) | Unchanged | 164 (59.6%) | 34 (34.3%) | 130 (73.9%) | **<0.001** |
| | Not employed prior to the pandemic | 67 (24.4%) | 52 (52.5%) | 15 (8.5%) | |
| | Change in employment due to the pandemic | 44 (16%) | 13 (13.1%) | 31 (17.6%) | |
| **Has COVID-19 impacted your housing situation?**, n (%) | No, I am in the same place | 267 (94.3%) | 99 (95.2%) | 168 (93.9%) | 0.43 |
| | No, but I still don't have a regular place | 5 (1.8%) | 3 (2.9%) | 2 (1.1%) | |
| | Yes, I had to move, but found a place to stay | 9 (3.2%) | 2 (1.9%) | 7 (3.9%) | |
| | Yes, I no longer have a place to stay | 2 (0.7%) | 0 (0%) | 2 (1.1%) | |
| **COVID-19 exposure and testing experiences** | | | | | |
| **Do you know, or think, that you have or previously had coronavirus?**, n (%) | Yes, currently have it | 1 (0.4%) | 0 (0%) | 1 (0.6%) | **0.0004** |
| | Yes, previously had it | 11 (3.9%) | 7 (6.7%) | 4 (2.2%) | |
| | No | 248 (87.6%) | 81 (77.9%) | 167 (93.3%) | |
| | Don't know | 23 (8.1%) | 16 (15.4%) | 7 (3.9%) | |
| **Ever tested for coronavirus[b]**, n (%) | Yes | 89 (31.4%) | 26 (25%) | 63 (35.2%) | 0.08 |

*(Continued)*

**Table 1.** (Continued)

| Characteristic | | Total (n = 283)* | Brooklyn, NY (n = 104) | New Haven, CT (n = 179) | p-value |
|---|---|---|---|---|---|
| **If tested for coronavirus, why did you get tested?[§§], n (%)** | I was exposed by a family member | 5 (6%) | 0 (0%) | 5 (8%) | **0.02** |
| | I was exposed by someone at work | 6 (7%) | 0 (0%) | 6 (10%) | |
| | I was exposed to someone else | 5 (6%) | 2 (8%) | 3 (5%) | |
| | Work recommended testing | 10 (11%) | 1 (4%) | 9 (14%) | |
| | The health department recommended testing | 11 (12%) | 1 (4%) | 10 (16%) | |
| | Other reason[d] | 52 (58%) | 22 (85%) | 30 (48%) | |
| **If tested for coronavirus, what was the outcome of the testing?[§§§], n (%)** | Positive | 6 (6.9%) | 2 (8%) | 4 (6.5%) | **0.04** |
| | Negative | 76 (87.4%) | 19 (76%) | 57 (91.9%) | |
| | Pending | 4 (4.6%) | 3 (12%) | 1 (1.6%) | |
| | Indeterminate | 1 (1.1%) | 1 (4%) | 0 (0%) | |
| **If not tested for coronavirus, what was the reason?[§§§§], n (%)** | My healthcare provider did not think it was necessary | 4 (2.1%) | 1 (1.3%) | 3 (2.6%) | **<0.001** |
| | I did not know where to go | 10 (5.2%) | 9 (11.7%) | 1 (0.9%) | |
| | I was afraid of how I would be treated if I was COVID-19 positive | 159 (82.4%) | 48 (62.3%) | 111 (95.7%) | |
| | I did not think it was necessary | 20 (10.4%) | 19 (24.7%) | 1 (0.9%) | |

All variables summarized with frequency and % except where noted

IQR = interquartile range

Bold values indicate statistically significant results

* Numbers do not add up to 100% due to small amounts of missing data

[a] >50 copies/mL

[b] The survey question was "Have you been tested for the coronavirus?" and responses were dichotomized as "yes" ("yes") vs. "no" ("in process", "unable to get testing", or "no")

[c] The survey question was "How much has COVID-19 impacted your ability to get or pay for food?" and responses were dichotomized as "yes" ("a little", "some", or "a lot") vs. "no" ("not at all")

[d] Other reasons (n = 52) included required hospital protocol (n = 18), for own knowledge or reassurance (n = 11), experiencing COVID-19 symptoms (n = 8), having concerns due to HIV status and/or other comorbidity (n = 6), protecting self and others (n = 4), recommended by healthcare provider (n = 2), accessible COVID-19 testing (n = 2), and no response (n = 1)

[§] n = 277

[§§] n = 89

[§§§] n = 87

[§§§§] n = 193

a regular place (1.8%), or no longer had a place to stay (0.7%). However, the majority of participants (94.3%, n = 267) remained in the same residence during the COVID-19 pandemic.

Regarding patient participants' perceptions of prior SARS-CoV-2 infection, most participants believed that they did not have SARS-CoV-2 (87.6%, n = 248). Among these participants, more patients receiving care in New Haven, CT than those in Brooklyn, NY believed that they did not have SARS-CoV-2 (93.3% vs. 77.9%, p = 0.0004). Among participants who had tested for SARS-CoV-2 (n = 89), the majority tested negative (n = 76, 87.4%) while a small number reported positive test results (n = 6, 6.9%). Among those who were tested for SARS-CoV-2 (n = 89), the most common reasons for testing were a requirement during a hospital visit (n = 18), a recommendation by the health department (n = 11), and for own knowledge or reassurance (n = 11). Among those who did not get tested for SARS-CoV-2 (n = 193), the most common reason was being afraid of how they would be treated if found positive (n = 159, 82.4%) with more participants engaged in care in New Haven, CT endorsing this concern than those in Brooklyn, NY (95.7% vs. 62.3%, p < 0.001).

**Patient participants' food insecurity experiences.** Of all surveyed patient participants, nearly a third (n = 87, 30.7%) reported that they experienced food insecurity during the first wave of the COVID-19 pandemic. Participants who received care in Brooklyn, NY were more likely to report experiencing food insecurity than participants in New Haven, CT (49% vs. 20.1%, p < 0.001). In a multivariable regression model (Fig 1), higher odds of experiencing food insecurity were associated with receiving care in Brooklyn, NY compared to New Haven, CT (OR 2.26 [CI 1.07 to 4.79], p = 0.03) as well as reporting some or a lot of changes in daily routine due to the pandemic (OR 2.38 [CI 1.19 to 4.76], p = 0.01). Compared with patient participants who indicated change or no change in their employment status due to the COVID-19 pandemic, the odds ratio of experiencing food insecurity was higher among participants reporting not working prior to the pandemic (OR 3.07 [CI 1.42 to 6.64]; p = 0.004). There was no significant difference in food insecurity experiences during the pandemic by age (based on a 1-year increase in age), gender, race, ethnicity, number of people in household, household income, presence of a detectable viral load, and recent engagement in HIV care (p value >0.05).

## Qualitative results

**Clinical staff characteristics.** Twenty-three clinical staff members from both sites participated in four focus groups, two at each site. Almost half (47.8%, n = 11) of the participating staff members were physicians; other participants included behavioral health providers (17.4%, n = 4), nurses (8.7%, n = 2), other clinical staff (17.4%, n = 4), a clinical pharmacist (4.3%, n = 1), and an advance practice practitioner (4.3%, n = 1).

**Focus group and free-text findings regarding factors affecting food insecurity and COVID-19-related experiences among PWH.** Analysis of the qualitative data comprising clinical staff focus group transcripts and patients' free-text responses showed thematic consistency, suggesting dominant theme saturation around multilevel factors

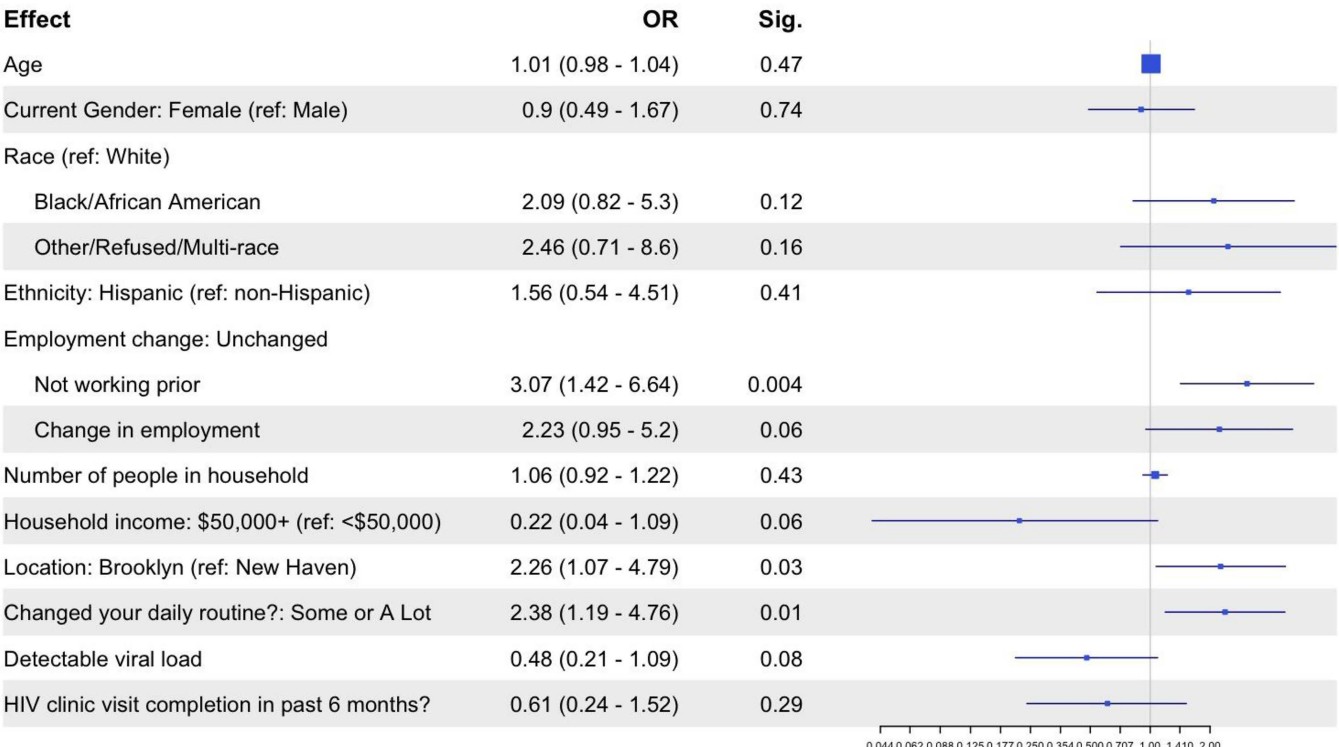

| Effect | OR | Sig. |
|---|---|---|
| Age | 1.01 (0.98 - 1.04) | 0.47 |
| Current Gender: Female (ref: Male) | 0.9 (0.49 - 1.67) | 0.74 |
| Race (ref: White) | | |
| Black/African American | 2.09 (0.82 - 5.3) | 0.12 |
| Other/Refused/Multi-race | 2.46 (0.71 - 8.6) | 0.16 |
| Ethnicity: Hispanic (ref: non-Hispanic) | 1.56 (0.54 - 4.51) | 0.41 |
| Employment change: Unchanged | | |
| Not working prior | 3.07 (1.42 - 6.64) | 0.004 |
| Change in employment | 2.23 (0.95 - 5.2) | 0.06 |
| Number of people in household | 1.06 (0.92 - 1.22) | 0.43 |
| Household income: $50,000+ (ref: <$50,000) | 0.22 (0.04 - 1.09) | 0.06 |
| Location: Brooklyn (ref: New Haven) | 2.26 (1.07 - 4.79) | 0.03 |
| Changed your daily routine?: Some or A Lot | 2.38 (1.19 - 4.76) | 0.01 |
| Detectable viral load | 0.48 (0.21 - 1.09) | 0.08 |
| HIV clinic visit completion in past 6 months? | 0.61 (0.24 - 1.52) | 0.29 |

**Fig 1. Multivariable Regression Model of Factors Associated with Food Insecurity Experiences among PWH.**

affecting food insecurity experiences among PWH during the COVID-19 pandemic that are parallel to the social-ecological model [39,40]. These factors were classified into the following levels: (1) *Individual or intrapersonal* factors encompassing themes that were grounded in the individual's thoughts, beliefs, or personal circumstances; (2) *Interpersonal* factors pertinent to participants' experiences with and reactions to or from other individuals, social networks, and support systems; and (3) *Community and structural* factors impacting an individual's experiences within the larger societal systems such as access to resources, social norms, and policies.

**Individual or intrapersonal factors.** On the individual or intrapersonal level, patient and staff participants described how mental health factors (e.g., anxiety) and physical function contributed to food insecurity experiences among PWH. At the beginning of the pandemic, *anxiety* about contracting SARS-CoV-2 prevented many people from leaving the house. Patient participants expressed their anxiety around leaving home in general as well as anxiety around obtaining food and groceries. Clinical staff also spoke about the emotional toll of food procurement for their patients, particularly among the *aging* population. Providing groceries for these patients seemed to alleviate food insecurity experiences and also anxiety surrounding going outside of their homes.

*"It's scary. I'm scared. I don't even go to the hallway. I don't go out. I don't do anything. I stay home. It's serious."* (Patient 70)

*"I don't go to the store, but when I do go, I'm very nervous and have a lot of anxiety. I'm scared about getting (SARS-CoV-2), you never know."* (Patient 62)

*"…Our patients are older, as you know. So many of them are older, they don't have access to food close by."* (Physician 1)

*"…The biggest issue is grocery shopping… (Patients) have had to venture out to go to the store and both the stress, the emotional stress of it and sort of the physical stress of it, you know, has been challenging."* (Physician 3)

*"We focused (our efforts to provide groceries) initially on patients over 60 because they seem to be more isolated and they had more trouble getting out... They were, I have to say, they were very overwhelmed by being able to not only get the groceries, but not have to go out."* (Behavioral Health Provider 1)

Many PWH also experienced *physical health* issues and comorbid conditions that might impede their efforts to obtain food. These barriers included musculoskeletal disorders, underlying cardio-pulmonary diseases, and other conditions that could affect mobility.

*"I have asthma, COPD, diabetes, and HIV, so I am very worried about catching the virus. I am worried that I will die because I have problems in my lungs."* (Patient 18)

*"People who don't normally walk very far are walking to the corner store and you know, having knee pain and back pain and everything else so that, that has been a challenge."* (Physician 3)

*"…We have surveyed the clients two different times formally about their food insecurity and in most cases it's moderate food insecurity and… not high need. For the patients, there might have been out of 120 people, one or two people with high need and that was associated with other issues like substance use, or co-occurring disorders, you know, untreated bipolar disorder, and insecure housing, too."* (Assistant Director of Care Coordination)

*"…A number of (patients) had other issues too, respiratory problems cardiac issues, and mobility issues…"* (Behavioral Health Provider 1)

Beyond physical and psychological barriers, clinical staff noticed that there were patients who might have experienced food insecurity during the COVID-19 pandemic as a result of being reluctant to utilize available resources and *feeling*

 

*guilty over accessing food support services*. Clinical staff described their efforts to assess PWH's basic needs and make services and resources available during the pandemic. Despite being eligible for these services, some patients simply refused to accept the referral or were cognizant about requesting only the items that they needed. Other patients felt guilty about accessing food resources and other services due to their *immigration status*, or due to fear of the Administration for Children's Services (ACS) and the hypothetical threat of children being removed from home.

*"I suddenly remembered somebody who I spoke to directly and who said he was, he had food insecurity and I gave him this information. And then I realized… he didn't take down the information, you know. He acted as if he was going to and I was very excited I had a resource for him, but he wasn't intending to follow up with it."* (Assistant Director of Care Coordination)

*"(Patients) were very particular about what they asked for in that everyone wanted to make sure in their own words that they didn't ask for too much, which I thought, these are not folks who had a lot to begin with. (Some) actually asked for two things, and I had to coax them into giving me a list because they said, 'I don't, I don't want to take away from anybody else…'"* (Behavioral Health Provider 1)

*"One of my clients, she's somebody who is undocumented, and even though she is eligible for some services like a pantry, she has a lot of guilt around utilizing services as someone who's not a taxpayer, so that's been a little bit difficult. She has still been able to get enough food, but I have the sense that she could benefit from, you know, additional resources."* (Behavioral Health Provider 4)

*"We noticed that even though they have food insecurity, they do not want the delivery of the food and I don't know if it's pride, I don't know if it's they don't want it to be singled out that they have those issues because they live in a building or in a neighborhood that they're not comfortable… 'Oh my god, I have COVID. Oh my god… I don't have food in my house. ACS is going to come and take my kids.' You know, there was a lot of worries on that area… We… (wanted to) make sure that everything is okay, the kids have food. Some patients for whatever reason they didn't want to show, just refused the referral. They refused to go to those places that the city opened to get the… cans and the food for that week."* (Assistant Director of Care Management)

**Interpersonal factors.** Patients described interpersonal factors that amplified the impact of the COVID-19 pandemic on their food insecurity experiences. An interpersonal factor reported by patient participants and clinical staff was related to PWH's significant *roles in the family* as a caregiver (e.g., parents with children or carers to older adults) or a breadwinner. Although a number of patients reported that they were not working prior to the pandemic, others continued to work outside the home in *essential work roles* to support and feed themselves and their families. These responsibilities often required physical proximity with other people and hampered the *ability to social distance* that could increase the risk of acquiring SARS-CoV-2 and subsequently, the risk of transmitting the virus to family members.

*"I care for an elderly parent and want to assure her safety."* (Patient 39)

*"A lot of my patients, particularly my female patients, are caregivers for other family members…and so they're not just doing grocery shopping for themselves…(they) have taken that upon themselves and putting their parents who's elderly with other comorbidities ahead of their own risk. So, they really feel like they have not a lot of choice but to… go out and do these things.*" (Physician 9)

*"It's not just them in their immediate family… you know. Older relatives that live near them, and (our patients are) helping to take care of them."* (Advanced Practice Provider)

*"I'm thinking of one particular patient that it's the patient, their parents, the parents' parents, and the parents' grandparents. So all of these generations in the same house and two of the family members did get COVID so managing that… And then, you know, one of the family members was having to go out was a home health aide and so then, you know, it's hard when you're living in poverty and you have a job and you need to continue working, right?"* (Behavioral Health Provider 4)

*"So I have a family, one of the spouses works in a nursing home and then comes home and there's six people, a couple are elderly… That raises a lot of concern about how do you… keep everyone safe in circumstances where one person's really going out and about quite a bit more than the others?"* (Behavioral Health Provider 2)

**Community and structural factors.** Staff and patients commented on extensive community and structural factors that contributed to food insecurity experiences during the first wave of the COVID-19 pandemic. Primary among these factors was the *increased demand for food services*, resulting in both reduced access to food and *variability in available food assistance options*. For some, the decreased access to food was a result of the overwhelming demand on organizations providing food services; for others, it resulted from the *higher cost of food*. Staff also reported that some patients declined referrals for food services as a result of perceived *HIV stigma*.

*"You know, different community-based organizations… their level to provide services… they really were overwhelmed at certain points, had to shut down at certain points."* (Assistant Director of Care Coordination)

*"We did have some trouble with one of our… resources… because at the beginning, it was just like chaos and they weren't processing referrals. They weren't answering the phones. It was just… very difficult."* (Behavioral Health Provider 4)

*"The price of food went up, now I have a few cents left and I used to have $40 left at the end of the month. Food prices have gone up."* (Patient 2)

*"Even though the resources were there, some patients opt to not get the services. It's like, it was a stigma about HIV."* (Assistant Director of Care Management)

Clinical staff members spoke of the use of online ordering and food delivery services such as Instacart to help with food access during the stay-at-home orders, and acknowledged the prohibitive costs of these services made them inaccessible to most patients. As a result, patients relying on food assistance services often had decreased or unpredictable *variability in the type of food* they were able to access.

*"I'm assuming that (Instacart) is not within the means of most of our patients to do that. And so, they have to go to the store physically and put themselves at potential risk."* (Physician 9)

*"Things like… Instacart or delivery are really not an option for them."* (Physician 3)

*"There are people… that are using the [city] delivery system and… the food is variable. Sometimes it's dry shelf stable food and sometimes it's… cooked chicken or hamburgers and things like that. So, in some cases, they're getting, you know, hot cooked meals.* (Assistant Director of Care Coordination)

Finally, patients and staff commented on the logistical barriers presented during the pandemic that contributed to food insecurity experiences among PWH. In particular, patients lacked the means to obtain food, as many did not have a personal vehicle or a safe *access to public transportation.* Additionally, when asked the question, "What has made it hard for you to practice social distancing?," 10 (3.5%) patient participants indicated, "I go to a food pantry/soup kitchen/food

pick-up location for food." To obtain groceries outside of the house, PWH placed themselves at risk of being exposed to SARS-CoV-2 when travelling to and shopping in grocery stores or other sites for food procurement. Both PWH and clinical staff spoke of *keeping social distance and wearing a mask following the COVID-19 guidelines* to reduce the risk of exposure, and how this was not always possible to do in the community including at food stores and in public transport.

> *"The issue of public transportation is an enormous one because people just don't have other ways of getting around, unless they have resources."* (Physician 8)

> *"I had to ask people to move in stores. People are anxious to get to the register and get back home, so they forget about social distancing."* (Patient 75)

> *"People are not respecting space, crowding at stores, not wearing masks."* (Patient 180)

> *"(Patients) were using old paper masks that were falling apart. They had no resources to go and buy another mask. They weren't sure where to get the masks."* (Behavioral Health Provider 1)

> *"Patients certainly tell me that they're respecting social distancing as best they can. But, you know, a lot of them have to, in order to go shopping, they have to take the bus, right? And sometimes on the bus you can social distance and sometimes you can't. I mean, sometimes in stores you can social distance and sometimes you can't. So they're doing the best they can, but it's not always ideal."* (Advance Practice Practitioner)

> *"What's interesting is, you know, the city's choice to not charge for the buses, which I think is wonderful in regard to the fact that the drivers are protected somewhat with this plastic sheet they have… to separate the driver from the main carriage of the car… But from what I've observed... the buses seem busier than ever. And that's really scary to me because yes... I think they're required to wear a mask, but people aren't wearing their masks correctly. And there are far too many people packed up on these buses like sardines and so that... that's unnecessary exposure, really."* (Behavioral Health Provider 4)

### Linkages between experiences of food insecurity and COVID-19 among PWH: A conceptual framework

Informed by findings from this study, we developed a framework to illustrate the linkages between experiences of food insecurity and COVID-19 among PWH (Fig 2), adapted from the work of Weiser et al. on the bidirectional links between food insecurity and HIV acquisition and disease severity at the individual, household, and community levels [13]. S1 Table outlines the sources of evidence emerged from the quantitative and qualitative data on social-ecological factors concerning food insecurity experiences, and pathways through which food insecurity experiences are linked with COVID-19 exposure and testing experiences among PWH.

Our quantitative and qualitative study findings generated some exploratory evidence to support the conceptual framework, demonstrating individual/intrapersonal factors that might impact food insecurity experiences among PWH during the COVID-19 pandemic, namely, age, gender, race, ethnicity, immigration status, household income, location, physical health, change of employment status, housing situation, and HIV care engagement. We also identified interpersonal factors that could affect food insecurity experiences in the context of COVID-19 pandemic, i.e., PWH's role in the family as a caregiver and/or a breadwinner, essential work role, and number of household members. Further, community and structural factors including increased demand of food assistance, rising food prices, perceived HIV stigma, and lack of access to public transportation, might influence the experiences of food insecurity among PWH in our study. In turn, food insecurity experiences might shape individual actions and health outcomes in PWH through nutritional, mental health, and behavioral pathways. Regarding the nutritional pathways, limited variability in the type of food accessible to PWH during the pandemic might lead to undernutrition that then could impact on disease progression in PWH. Through mental

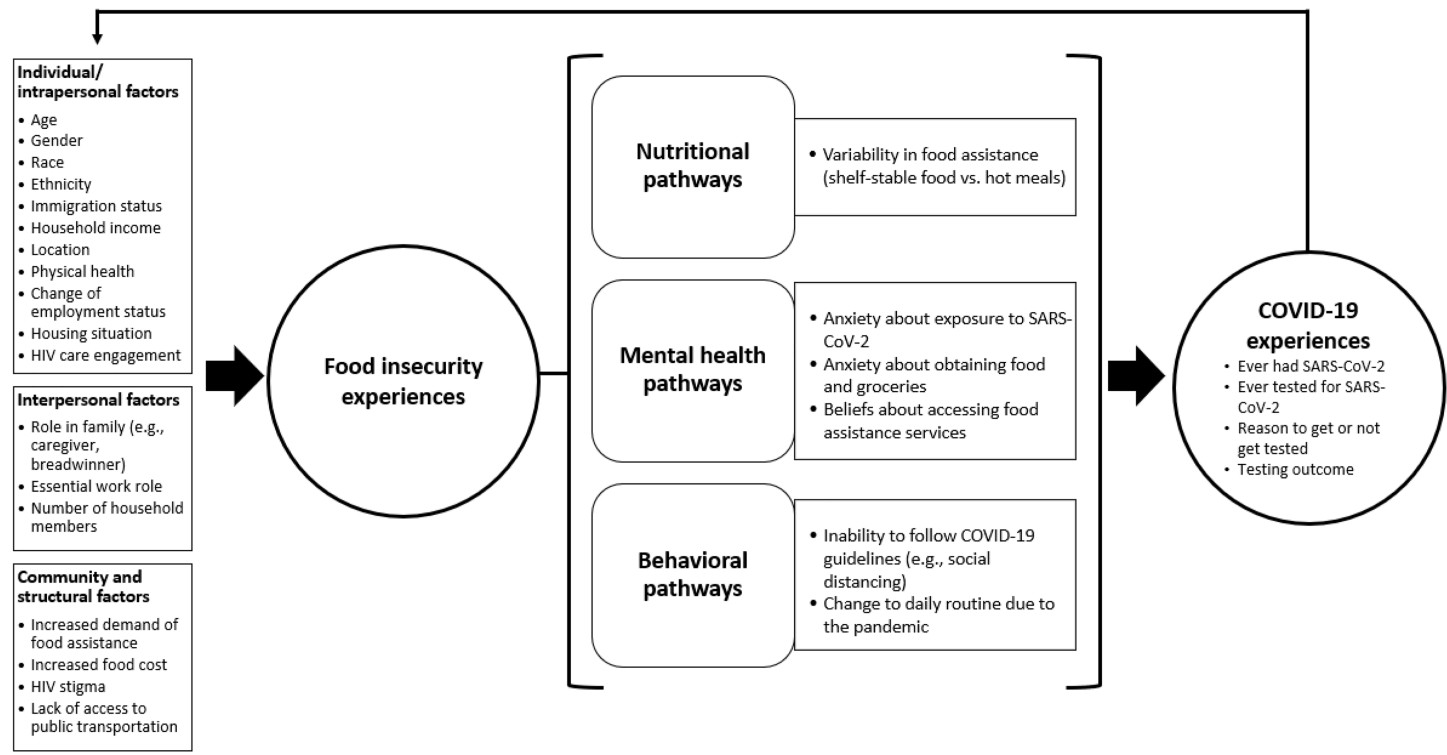

**Fig 2. A Conceptual Framework for the Linkages Between Experiences of Food Insecurity and COVID-19 among PWH.** This figure illustrates individual, interpersonal, and community and structural factors affecting the experiences of food insecurity among PWH in the study. These multilevel factors could then shape individual actions, health outcomes, and COVID-19 experiences among PWH through nutrition, mental health, and behavioral pathways.

health pathways, anxiety about the pandemic and obtaining food and groceries, and also beliefs about accessing food assistance services could contribute to food insecurity experiences. Lastly, change to daily routine and inability to follow COVID-19 guidelines such as social distancing were behavioral pathways that could increase the risk of SARS-COV-2 acquisition and adversely affect health outcomes in PWH during the pandemic.

## Discussion

To our knowledge, this is the first study engaging multiple stakeholders to explore the relationships between socio-demographic factors, HIV care engagement, and food insecurity experiences among PWH in the context of the COVID-19 pandemic. Our mixed methods study findings suggest that nearly a third of PWH participating in the study experienced some levels of perceived food insecurity during the first wave of the COVID-19 pandemic. We found some significant differences between patient populations receiving care in Brooklyn, NY versus New Haven, CT with patients at the Brooklyn, NY site being more likely to report experiencing food insecurity. We observed that self-reported food insecurity experiences among PWH were associated with receiving care in Brooklyn, NY, reporting not having employment prior to the COVID-19 pandemic, and undergoing some or a lot of changes in daily routine due to the pandemic. Beyond these quantitative findings, our qualitative results identified individual, interpersonal, and community and structural factors affecting the experiences of food insecurity among PWH in our study. These multi-level factors could then shape individual actions, health outcomes, and COVID-19 experiences among PWH through nutrition, mental health, and behavioral pathways.

Our results were consistent with studies documenting the impact of the COVID-19 pandemic and associated public health measures on PWH's ability to meet basic needs, including food [41,42]. In a mixed methods study of Black and Latino PWH in New York City also conducted around the early stage of the pandemic, only 8% of 96 participants indicated working full-time or part-time [42]. Among 84% of participants in the study who were identified as food-insecure, restricted travel and reduced food quality were cited as causes for food insecurity during the pandemic. Similarly, among men and women with HIV in Atlanta [41], 80% of 162 participants reported being unemployed and nearly 40% were unable to obtain food as a result of actions intended to reduce their risk of SARS-CoV-2 acquisition, including social distancing and avoidance of public transport. Together, these findings point to pre-existing challenges with food insecurity experiences for PWH that were exacerbated both by the public health response to the pandemic, and by the reduced availability of compensatory resources.

A unique aspect of our study was the development of a conceptual framework illustrating the linkages between experiences of food insecurity and COVID-19 among PWH. Early recognition of individual factors contributing to food insecurity experiences – such as age, physical health, and household income as shown in our sample of PWH who were older individuals of lower socioeconomic status with comorbid conditions – might have been critical for staff in rapidly addressing some of the underlying drivers of a patient's food shortage, and therefore in helping patients to navigate food access during the pandemic. Other factors identified by patient and staff participants highlighted in the framework such as increased demand of food assistance and lack of access to public transportation, underscore the complex and changing community-level drivers of food insecurity experiences in the midst of a public health crisis, as well as the broader structural variables exacerbating health disparities beyond the COVID-19 pandemic context. Notably, decreased engagement in HIV care was also associated with food insecurity experiences: those patients who were poorly connected to their HIV care were likely less able to partner with their medical team in addressing other relevant individual- and community-level factors contributing to their food insecurity experiences. The proposed conceptual framework, while not exhaustive, provides a starting point supported by exploratory evidence for identifying the multitude of factors that may affect food insecurity experiences among PWH. The framework may also inform understanding of the potential role that HIV clinics can play in both evaluating and mitigating food insecurity experiences as a modifiable social determinant of health. Furthermore, the framework can serve as a model to guide future investigation in addressing complex drivers of food insecurity among PWH and coordinating potential pandemic responses.

The present study has several strengths. Our study adds to the literature by documenting multilevel factors affecting food insecurity experiences among PWH during the COVID-19 pandemic and identifying potential practice and research implications. Our conceptual framework that highlights the links between experiences of food insecurity and COVID-19 can guide future research and programs to understand the social-ecological factors that influence food insecurity among PWH, and to inform strategic interventions to address both food insecurity and HIV-related health outcomes. For example, conducting qualitative research with PWH can provide insights into the lived experiences of PWH related to food insecurity and the impact on their health [43]. Future intervention research efforts could evaluate the effectiveness and sustainability of incorporating food and nutritional support interventions for PWH as part of their HIV care plan [44,45]. Providing PWH affected by food insecurity with food assistance, either in the form of vouchers or cash transfer, has shown promising results in improving medication adherence and HIV care engagement when tailored to the local needs and settings [46]. Regarding practice implications in HIV care settings, establishing regular screening for food insecurity [47] and implementing strategies to better communicate about available food assistance programs and other resources can help alleviate stigma or shame around accessing these services [48].

Our study had several limitations. First, our modest sample size of English-speaking PWH (most of whom had documented tobacco use) and clinical staff in the U.S. Northeast, potentially limit our ability to compare the experiences of PWH and staff in different locations, PWH without tobacco use, and individuals who did not primarily speak English. Second, our study evaluated a range of COVID-19 pandemic experiences among PWH through a survey and did not focus specifically on

the impacts of the pandemic on social determinants of health. Our qualitative analysis was primarily informed by the clinical staff's perspectives; as a result, we were limited in our ability to further explore important nuances regarding food insecurity experiences among PWH. Third, although we employed and adapted an existing survey to assess the impact of COVID-19 pandemic on HIV care, we did not incorporate a validated scale to measure food insecurity, such as the one used by the U.S. Census Bureau [49,50]. Thus, our study captured perceived experiences of food insecurity among PWH, rather than actual food insecurity status. Fourth, we acknowledge that there is potential for non-response bias in our study due to the significant rate of unsuccessful telephone outreach to eligible patient participants. Since we could not obtain consent from unreached individuals or those who declined participation, we were unable to compare the sociodemographic factors and clinical characteristics among respondents and non-respondents or determine in which direction the response rate might bias the results. Finally, given our modest sample size, we may not have been sufficiently powered to parse statistically significant associations between measured variables and food insecurity experiences as primary outcomes of interest.

## Conclusions

Our study provided insights into PWH's experiences related to food insecurity in the context of the COVID-19 pandemic. Efforts to address food insecurity among PWH should include both research and programmatic strategies focused on screening for food needs in HIV care settings, communicating about food assistance resources and providing referrals, and implementing evidence-based interventions that can improve food security and nutrition.

## Supporting information

**S1 Table. Sources of Exploratory Evidence Emerged from the Study Data on Social-Ecological Factors and Pathways Shaping Experiences of Food Insecurity and COVID-19 among PWH.**
(XLSX)

**S1 File. Survey Dataset.**
(XLSX)

**S2 File. Qualitative Codebook.**
(DOCX)

## Acknowledgments

The authors would like to thank Renee Capasso, Tequetta Valeriano, Katarzyna Sims, Elena Sullivan, Stephanie Salas, J. Morgan Jones, Indumathi Dhakshinamurthy, and Jona Tanguay for their efforts in data collection; Elizabeth Porter for her efforts in coordinating study activities; Christopher Cole for his input on survey design; and Eliott Wang for data entry.

## Author contributions

**Conceptualization:** Dini Harsono, Kelsey Sklar, Michael Silver, Mayange Frederick, Merceditas Villanueva, E. Jennifer Edelman, Jessica E. Yager.

**Data curation:** Dini Harsono, Kelsey Sklar, Michael Silver, Mayange Frederick, E. Jennifer Edelman, Jessica E. Yager.

**Formal analysis:** Dini Harsono, Kelsey Sklar, Michael Silver, E. Jennifer Edelman, Jessica E. Yager.

**Funding acquisition:** E. Jennifer Edelman.

**Investigation:** Dini Harsono, Mayange Frederick, E. Jennifer Edelman, Jessica E. Yager.

**Methodology:** Dini Harsono, Kelsey Sklar, Michael Silver, Mayange Frederick, Merceditas Villanueva, E. Jennifer Edelman, Jessica E. Yager.

**Project administration:** Mayange Frederick.

**Software:** Michael Silver.

**Supervision:** E. Jennifer Edelman, Jessica E. Yager.

**Validation:** Dini Harsono, Michael Silver.

**Visualization:** Dini Harsono, Michael Silver.

**Writing – original draft:** Kelsey Sklar, Michael Silver, Mayange Frederick, Jessica E. Yager.

**Writing – review & editing:** Dini Harsono, Kelsey Sklar, Michael Silver, Mayange Frederick, Merceditas Villanueva, E. Jennifer Edelman, Jessica E. Yager.

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
