## [Decision Letter · Decision Letter 0]

11 Nov 2025

Dear Dr. Harsono,

Thank you for submitting your manuscript to PLOS ONE. After careful consideration, we feel that it has merit but does not fully meet PLOS ONE’s publication criteria as it currently stands. Therefore, we invite you to submit a revised version of the manuscript that addresses the points raised during the review process.

We look forward to receiving your revised manuscript.

Kind regards,

Sungwoo Lim, DrPH

Academic Editor

PLOS ONE

[This work was supported by funding from the National Institute of Mental Health (grant #P30MH062294) and the National Cancer Institute (grant #R01CA243910) of the National Institutes of Health. The content is solely the responsibility of the authors and does not necessarily represent the official views of the Center for Interdisciplinary Research on AIDS or the National Institutes of Health.].

3. In the online submission form, you indicated that [A de-identified dataset and codebooks are available upon request to the corresponding author, and with approval of study Principal Investigators. Requests for restricted use of a de-identified dataset will require a data use agreement that outlines the terms and conditions that may include limitations on the use of data, obligations to safeguard the data, and privacy rights that are associated with transfers of confidential or protected data. The data use agreement is managed by the Yale University's Office of Sponsored Projects.].

Additional Editor Comments (if provided):

Reviewers' comments:

Reviewer's Responses to Questions

**Comments to the Author**

1. Is the manuscript technically sound, and do the data support the conclusions?

Reviewer #1: Yes

2. Has the statistical analysis been performed appropriately and rigorously?

Reviewer #1: No

3. Have the authors made all data underlying the findings in their manuscript fully available?

Reviewer #1: Yes

4. Is the manuscript presented in an intelligible fashion and written in standard English?

Reviewer #1: No

Reviewer #1: General Assessment

The manuscript addresses an important and timely topic by examining food insecurity among people with HIV (PWH) during the COVID-19 pandemic using a mixed-methods approach. The study provides valuable insights into multilevel drivers of food insecurity and integrates both quantitative and qualitative evidence from two urban clinics in the northeastern United States.

However, the paper requires substantial revisions to improve methodological transparency, data validity, and overall clarity. Several critical aspects of sampling, measurement, statistical analysis, and qualitative reporting need strengthening before the paper can meet PLOS ONE standards for rigor and reproducibility.

1. Sampling, Inclusion Criteria, and Generalizability

The study draws participants from an ongoing research collaboration on tobacco use among PWH, with recruitment based on documented tobacco use in the electronic medical record. This sampling frame introduces potential selection bias, as tobacco users may differ from non-users in socioeconomic, behavioral, and health characteristics that are also associated with food insecurity.

Recommendation: Provide a detailed explanation of inclusion and exclusion criteria. Explicitly state whether all participants were current tobacco users. Discuss the implications of this sampling design for the generalizability of results, and include this limitation in the Discussion section.

2. Response Rate and Non-Response Bias

Of 719 individuals contacted, only 283 completed the survey (~39% response rate). This relatively low participation rate raises concerns about non-response bias, particularly if respondents differ systematically from those who did not participate.

Recommendation: If possible, compare key demographic or clinical characteristics between respondents and non-respondents using available electronic medical record data (e.g., age, gender, race, CD4 count, last clinic visit). If such data are unavailable, at least discuss the direction and magnitude of possible bias and its potential effect on prevalence estimates and regression outcomes.

3. Measurement of Food Insecurity

Food insecurity was assessed using a single self-reported item about the perceived impact of COVID-19 on participants’ ability to obtain or pay for food. While this item provides contextual information, it does not represent a validated measure such as the USDA Household Food Security Survey Module or the HFIAS. Consequently, prevalence estimates and comparisons with other studies may be unreliable.

Recommendation:

• Justify the choice of a single-item measure and acknowledge its limitations.

• Conduct sensitivity analyses using alternative cut-offs if applicable.

• Expand the Discussion to note that the measure captures perceived rather than experienced food insecurity.

4. Study Design and Causality

Because the study uses a cross-sectional design, it cannot establish causal relationships between COVID-related disruptions and food insecurity. The conceptual model presented in the Discussion implies bidirectional relationships, but these are not empirically tested.

Recommendation: Clearly distinguish between observed associations and hypothesized causal mechanisms. Rephrase causal language (e.g., “led to,” “caused”) into non-causal wording such as “was associated with.” Include this clarification in both the Abstract and Discussion.

5. Statistical Analysis and Model Transparency

The description of the statistical procedures lacks detail. It is unclear how variables were selected for the multivariable model, whether multicollinearity was assessed, how missing data were handled, and whether model assumptions were tested. The manuscript also does not provide full regression outputs (odds ratios, 95% confidence intervals, p-values for all predictors).

Recommendation:

• Present a complete multivariable logistic regression table.

• Specify the model-building strategy (a priori, stepwise, or significance-based).

• Report diagnostic statistics (e.g., VIF for multicollinearity, Hosmer–Lemeshow test, AUC).

• Describe how missing data were treated (listwise deletion, imputation, etc.).

• Conduct sensitivity analyses to test robustness (e.g., stratified by clinic site or alternate food insecurity thresholds).

6. Handling of Missing Data and Data Availability

Although the manuscript mentions minor missing data, there is no clear statement on how these were managed. Furthermore, PLOS ONE requires a transparent Data Availability Statement. The current version only states that data are “available upon request,” which does not comply with journal policy.

Recommendation:

• Describe how missing values were handled analytically.

• Deposit de-identified data and analysis code in a recognized repository (e.g., Dryad, Zenodo) or explain ethical restrictions preventing public release.

• Provide the repository link or DOI in the Data Availability section.

7. Ethical Approval and Informed Consent

The manuscript indicates approval by the Yale and SUNY Downstate Institutional Review Boards and that participants gave verbal consent, but the protocol approval numbers are not provided.

Recommendation:

Include IRB approval numbers for each institution and explain why verbal consent was used (e.g., due to remote data collection during the pandemic). Clarify how confidentiality and data protection were ensured during telephone interviews.

8. Funding Disclosure and Role of the Sponsor

Funding is acknowledged from NIH grants (P30MH062294; R01CA243910), but it is unclear which authors received these awards and whether the funders had any role in the study’s design, analysis, or publication decision.

Recommendation:

Provide full grant numbers, specify award recipients, and explicitly state the funders’ role in accordance with PLOS ONE disclosure requirements.

9. Qualitative Analysis: Rigor and Integration

The qualitative component lacks sufficient methodological detail. The process for coding, the number of coders, how inter-coder reliability or consensus was achieved, and whether data saturation was reached are not clearly described. The integration between quantitative and qualitative results is also limited.

Recommendation:

• Elaborate on the coding framework, analytical approach (inductive/deductive), and verification steps.

• Include exemplar quotations supporting each theme in a table or appendix.

• Explain how qualitative and quantitative findings were triangulated to construct the conceptual model.

10. Results Presentation and Figures/Tables

Tables and figures require additional clarity. Denominators, missing data counts, and confidence intervals should be consistently reported. The conceptual framework figure should include clear labeling and be explicitly referenced in the text.

Recommendation:

• Ensure all tables include n values, percentages, and missing cases.

• Add footnotes describing statistical tests used.

• Improve the figure’s readability and provide a descriptive caption explaining each component.

11. Writing Quality and Structure

While the paper is generally well organized, some sentences are awkwardly phrased or repetitive. The manuscript would benefit from professional English language editing to improve flow, grammar, and precision.

Recommendation:

• Revise for conciseness and clarity, especially in the Abstract and Discussion.

• Consider editing by a native English-speaking academic editor to ensure alignment with PLOS ONE style.

Summary Recommendation

The study contributes important evidence about the intersection of HIV, food insecurity, and pandemic-related stressors. However, before publication, the authors must substantially enhance methodological transparency and analytic rigor.

If the above revisions are adequately addressed—particularly concerning sampling clarification, statistical reporting, data availability, and ethical documentation—the manuscript could make a valuable contribution to the literature on health inequities and food security among vulnerable population.

**Do you want your identity to be public for this peer review?** For information about this choice, including consent withdrawal, please see our Privacy Policy

Reviewer #1: No

---

## [Author Response · Author response to Decision Letter 1]

23 Dec 2025

December 22, 2025

Response to the Editor and Reviewer

We thank the Editor and Reviewer for the consideration and thoughtful comments on our manuscript, PONE-D-25-41950: “Food insecurity and COVID-19-related experiences among people with HIV: A mixed methods analysis and conceptual framework.” We have addressed the concerns raised by the Editor and Reviewer, and we believe that the incorporated changes have improved the rigor and impact of our manuscript. We outlined our point-by-point responses to the comments below. Along with this response letter, we submitted a marked up copy of our revised manuscript with track changes as well as an unmarked version of the revised paper.

EDITOR’S COMMENTS

1. Please ensure that your manuscript meets PLOS ONE's style requirements, including those

for file naming. The PLOS ONE style templates can be found at

Response #1. We have reviewed PLOS ONE’s formatting guidelines and updated the manuscript accordingly.

[This work was supported by funding from the National Institute of Mental Health (grant #P30MH062294) and the National Cancer Institute (grant #R01CA243910) of the National Institutes of Health. The content is solely the responsibility of the authors and does not necessarily represent the official views of the Center for Interdisciplinary Research on AIDS or the National Institutes of Health.].

Response #2. We have included a revised funding statement in the cover letter of this resubmission.

Revised funding statement in cover letter: This work was supported by funding from the National Institute of Mental Health (grant #P30MH062294, P30 grant PI: Kershaw, pilot study grant PI: Edelman) and the National Cancer Institute (grant #R01CA243910, R01 grant PIs: Edelman and Bernstein) of the National Institutes of Health. The content is solely the responsibility of the authors and does not necessarily represent the official views of the Center for Interdisciplinary Research on AIDS or the National Institutes of Health. The funders had no role in study design, data collection and analysis, decision to publish, or preparation of the manuscript.

3. In the online submission form, you indicated that [A de-identified dataset and codebooks are available upon request to the corresponding author, and with approval of study Principal Investigators. Requests for restricted use of a de-identified dataset will require a data use agreement that outlines the terms and conditions that may include limitations on the use of data, obligations to safeguard the data, and privacy rights that are associated with transfers of confidential or protected data. The data use agreement is managed by the Yale University's Office of Sponsored Projects.].

Response #3. The survey data and qualitative codebook associated with this manuscript have been submitted as supplementary materials.

Response #4. The Reviewer’s comments included a recommendation to cite a validated food insecurity scale developed by the U.S. Department of Agriculture (USDA). We have added this citation to the discussion section and references.

REVIEWER’S COMMENTS

Reviewer #1: General Assessment

The manuscript addresses an important and timely topic by examining food insecurity among people with HIV (PWH) during the COVID-19 pandemic using a mixed-methods approach. The study provides valuable insights into multilevel drivers of food insecurity and integrates both quantitative and qualitative evidence from two urban clinics in the northeastern United States.

However, the paper requires substantial revisions to improve methodological transparency, data validity, and overall clarity. Several critical aspects of sampling, measurement, statistical analysis, and qualitative reporting need strengthening before the paper can meet PLOS ONE standards for rigor and reproducibility.

Response. Thank you for taking the time to carefully review our manuscript and share your feedback, and for endorsing the importance of the topic.

1. Sampling, Inclusion Criteria, and Generalizability

The study draws participants from an ongoing research collaboration on tobacco use among PWH, with recruitment based on documented tobacco use in the electronic medical record. This sampling frame introduces potential selection bias, as tobacco users may differ from non-users in socioeconomic, behavioral, and health characteristics that are also associated with food insecurity.

Recommendation: Provide a detailed explanation of inclusion and exclusion criteria. Explicitly state whether all participants were current tobacco users. Discuss the implications of this sampling design for the generalizability of results, and include this limitation in the Discussion section.

Response #1. We recognize that our sampling approach that actively recruited PWH with current tobacco use as a limitation, and accordingly have added text reflecting this to the discussion section.

Page 28, lines 601-604: First, our modest sample size of English-speaking PWH (most of whom had documented tobacco use) and clinical staff in the U.S. Northeast, potentially limit our ability to compare the experiences of PWH and staff in different locations, PWH without tobacco use, and individuals who did not primarily speak English.

2. Response Rate and Non-Response Bias

Of 719 individuals contacted, only 283 completed the survey (~39% response rate). This relatively low participation rate raises concerns about non-response bias, particularly if respondents differ systematically from those who did not participate.

Recommendation: If possible, compare key demographic or clinical characteristics between respondents and non-respondents using available electronic medical record data (e.g., age, gender, race, CD4 count, last clinic visit). If such data are unavailable, at least discuss the direction and magnitude of possible bias and its potential effect on prevalence estimates and regression outcomes.

Response #2. We acknowledge that there is potential for non-response bias in our study. A large number of non-respondents (289 out of 719) were individuals whom our study team members were unable to reach by phone despite multiple attempts. Of those patients successfully contacted (430 out of 719), 34.1% (n=147) declined participation.

Because we were not able to gather data on individuals who did not respond and those who declined participation, we can only hypothesize about potential differences between respondents and non-respondents, and about potential bias introduced by the response rate. We have updated the relevant sections in the methods and added clarifying text to the discussion as a limitation in our study.

Pages 9-10, lines 201-204: We outreached 719 individuals out of a random sample of 755 patients who met the eligibility criteria; 40.2% (n=289) were not successfully contacted by telephone despite multiple attempts. A total of 283 PWH (65.8% of those who were successfully outreached) agreed to participate in the survey, yielding a response rate of 39.4%.

Pages 28-29, lines 612-616: Fourth, we acknowledge that there is potential for non-response bias in our study due to the significant rate of unsuccessful telephone outreach to eligible patient participants. Since we could not obtain consent from unreached individuals or those who declined participation, we were unable to compare the sociodemographic factors and clinical characteristics among respondents and non-respondents or determine in which direction the response rate might bias the results.

3. Measurement of Food Insecurity

Food insecurity was assessed using a single self-reported item about the perceived impact of COVID-19 on participants’ ability to obtain or pay for food. While this item provides contextual information, it does not represent a validated measure such as the USDA Household Food Security Survey Module or the HFIAS. Consequently, prevalence estimates and comparisons with other studies may be unreliable.

Recommendation:

• Justify the choice of a single-item measure and acknowledge its limitations.

• Conduct sensitivity analyses using alternative cut-offs if applicable.

• Expand the Discussion to note that the measure captures perceived rather than experienced food insecurity.

Response #3. Thank you for this comment. Our study was focused on evaluating the impacts of COVID-19 pandemic on HIV care and lived experiences among PWH; food security experiences were therefore assessed as one of the outcomes of interest rather than as a standalone outcome. Although we adapted an existing survey assessing COVID-19 experiences among adults with chronic conditions, we did not employ a validated scale to measure food insecurity. As a result, prevalence estimates of food insecurity experiences should not be interpreted as actual prevalence of food insecurity that can be compared with other studies using validated measures of food insecurity. Indeed, we noted this as a limitation in our study in the discussion section. We have included a citation for the USDA Household Food Security Survey Module recommended by the reviewer in the discussion section.

Page 28, lines 608-610: Third, although we employed and adapted an existing survey to assess the impact of COVID-19 pandemic on HIV care, we did not incorporate a validated scale to measure food insecurity, such as the one used by the U.S. Census Bureau [48, 49].

Further, we have updated the relevant sections in the results and added text to the discussion to clarify that our study captured perceived experiences of food insecurity, rather than actual food insecurity status.

Page 26, lines 544-547: We observed that self-reported food insecurity experiences among PWH were associated with receiving care in Brooklyn, NY, reporting not having employment prior to the COVID-19 pandemic, and undergoing some or a lot of changes in daily routine due to the pandemic.

Page 28, lines 611-612: Thus, our study captured perceived experiences of food insecurity among PWH, rather than actual food insecurity status.

We did not conduct sensitivity analyses due to our modest sample size.

4. Study Design and Causality

Because the study uses a cross-sectional design, it cannot establish causal relationships between COVID-related disruptions and food insecurity. The conceptual model presented in the Discussion implies bidirectional relationships, but these are not empirically tested.

Recommendation: Clearly distinguish between observed associations and hypothesized causal mechanisms. Rephrase causal language (e.g., “led to,” “caused”) into non-causal wording such as “was associated with.” Include this clarification in both the Abstract and Discussion.

Response #4. Thank you for this comment. We have addressed this by revising the causal language throughout our manuscript to accurately reflect the exploratory nature of our current study.

We have also clarified that the conceptual framework was not exhaustive, but rather designed as an exploratory framework informed by the study findings. To better contextualize our findings and processes in developing the framework, we have submitted a supplementary table outlining the sources of exploratory evidence emerged from the study data on social-ecological factors and pathways through which food insecurity experiences are linked with COVID-19 experiences among PWH.

Page 24, lines 510-513: Supplementary Table S1 outlines the sources of evidence emerged from the quantitative and qualitative data on social-ecological factors concerning food insecurity experiences, and pathways through which food insecurity experiences are linked with COVID-19 exposure and testing experiences among PWH.

Page 27, lines 577-583: The proposed conceptual framework, while not exhaustive, provides a starting point supported by exploratory evidence for identifying the multitude of factors that may affect food insecurity experiences among PWH. The framework may also inform understanding of the potential role that HIV clinics can play in both evaluating and mitigating food insecurity experiences as a modifiable social determinant of health. Furthermore, the framework can serve as a model to guide future investigation in addressing complex drivers of food insecurity among PWH and coordinating potential pandemic responses.

5. Statistical Analysis and Model Transparency

The description of the statistical procedures lacks detail. It is unclear how variables were selected for the multivariable model, whether multicollinearity was assessed, how missing data were handled, and whether model assumptions were tested. The manuscript also does not provide full regression outputs (odds ratios, 95% confidence intervals, p-values for all predictors).

Recommendation:

• Present a complete multivariable logistic regression table.

• Specify the model-building strategy (a priori, stepwise, or significance-based).

• Report diagnostic statistics (e.g., VIF for multicollinearity, Hosmer–Lemeshow test, AUC).

• Describe how missing data were treated (listwise deletion, imputation, etc.).

• Conduct sensitivity analyses to test robustness (e.g., stratified by clinic site or alternate food insecurity thresholds).

Response #5. Thank you for the suggestions regarding the statistical analyses. As described in the methods section, we included all variables that were statistically significant in the univariable analyses (at the p = 0.05 level) and demographic factors that we thought were important to control for in the multivariable model. We did not include the table for the regression model as we felt it was redundant with the forest plot (Fig 1). We did not assess multicollinearity as the model itself is more illustrative of the relationship with food insecurity, rather than prescriptive.

In prior iterations of the analyses, we looked at different thresholds for food insecurity experiences. At first, we compared patient participants who reported whether the COVID-19 pandemic impacted their ability to get or pay for food by grouping responses as “not at all/a little” versus “some/a lot”. We then compared dichotomized responses of “not at all” versus “a little/some/a lot” to the same question. We settled on the latter definition to be consistent with prior research being conducted in this topic to capture any food insecurity experiences among our study participants. We found that there were very few differences in the prediction between the two analytical approaches.

Lastly, we did not perform a stratified analysis by site, as we were already underpowered and therefore did no

---

## [Decision Letter · Decision Letter 1]

28 Jan 2026

Food insecurity and COVID-19-related experiences among people with HIV: A mixed methods analysis and conceptual framework

PONE-D-25-41950R1

Dear Dr. Harsono,

We’re pleased to inform you that your manuscript has been judged scientifically suitable for publication and will be formally accepted for publication once it meets all outstanding technical requirements.

Kind regards,

Sungwoo Lim, DrPH

Academic Editor

PLOS One

Additional Editor Comments (optional):

Reviewers' comments:

Reviewer's Responses to Questions

**Comments to the Author**

Reviewer #1: All comments have been addressed

2. Is the manuscript technically sound, and do the data support the conclusions?

Reviewer #1: Partly

3. Has the statistical analysis been performed appropriately and rigorously?

Reviewer #1: Yes

4. Have the authors made all data underlying the findings in their manuscript fully available?

Reviewer #1: Yes

5. Is the manuscript presented in an intelligible fashion and written in standard English?

Reviewer #1: Yes

Reviewer #1: (No Response)

**Do you want your identity to be public for this peer review?** For information about this choice, including consent withdrawal, please see our Privacy Policy

Reviewer #1: No

---

## [Editor Report · Acceptance letter]

PONE-D-25-41950R1

PLOS One

Dear Dr. Harsono,

I'm pleased to inform you that your manuscript has been deemed suitable for publication in PLOS One. Congratulations! Your manuscript is now being handed over to our production team.

Kind regards,

on behalf of

Dr. Sungwoo Lim

Academic Editor

PLOS One